# New Fertilizer Strategies Combining Manure and Urea for Improved Rice Growth in Mozambique

**Fátima Ismael** [1,2,*], **Alexis Ndayiragije** [3] **and David Fangueiro** [4,*]

1 Faculty of Agrarian Sciences, Lúrio University, Department of Rural Development, Campus Universitário de Unango, Sanga District 3302, Mozambique
2 Nova School of Business and Economics, Universidade Nova de Lisboa, Campus de Carcavelos, Rua da Holanda Street 1, 2775-045 Lisbon, Portugal
3 International Rice Research Institute-Eastern and Southern Africa Hub, Maputo 2100, Mozambique; a.ndayiragije@irri.org
4 Linking Landscape, Environment Agriculture and Food (LEAF), Instituto Superior de Agronomia (ISA), Universidade de Lisboa, Tapada de Ajuda, 1349-017 Lisbon, Portugal
* Correspondence: fatima786.ismael@gmail.com (F.I.); dfangueiro@isa.ulisboa.pt (D.F.)

**Abstract:** The cost of chemical fertilizers is increasing and becoming unaffordable for smallholders in Africa. The present study aimed to assess the impact of combined fertilization strategies using urea and animal manure (beef cattle manure and poultry litter manure) on rice yield and nutrient uptake. For this, a field experiment was carried out on a loam sandy soil in the Chókwè Irrigation Scheme. We set seven treatments in a Randomized Complete Block Design (RCBD), namely: T0: no fertilizer, T1: 100% urea, T2: 100% beef cattle manure, T3: 100% poultry litter, T4: 50% urea + 50% beef cattle manure, T5: 50% urea + 50% poultry litter and T6: 40% urea + 30% beef cattle manure + 30% poultry litter, replicated four times each. All treatments, except T0, received an amount of nitrogen (N) equivalent to 100 kgN·ha$^{-1}$. Results revealed that the highest yield grain (425 g·m$^{-2}$), plant height (115 cm), number of tillers (18) and thousand-grain weight (34g) were observed in treatments combining urea with manure (T4, T5 and T6) indicating that N supply in the mixture (urea + manure) is more efficient than in isolated applications of N (T1, T2 and T3). The data obtained in this study suggest that a combination of fertilizers (T6) lead to competitive yields and is thus recommended for best soil management practices.

**Keywords:** organic fertilizer; chemical fertilizer; Rice (*Oryza sativa* L.); nutrient uptake

## 1. Introduction

Rice plays a major role in global food security and is becoming the staple food of more than half of the world population, providing principal nutrients in the different continents, particularly in Asia, Latin America and Africa [1]. In Africa, rice production is below growing consumer demand coupled to increases in population, rapid urbanization and changing consumer behavior [2]. The African continent accounts for only 4.8% of the world rice production, and despite its potential, with 130 million hectares of arable soil, only 8% is currently under cultivation [3]. The average rice yield in Africa (2 Mg·ha$^{-1}$) is much lower than in Asian countries, e.g., Indonesia, Bangladesh, Vietnam and Myanmar, which produce 7 Mg·ha$^{-1}$ [4].Rice plays a major role in global food security and is becoming the staple food of more than half of the world population, providing principal nutrients in the different continents, particularly in Asia, Latin America, and Africa [1]. In Africa, rice production is below growing consumer demand coupled to increases in population, rapid urbanization and changing consumer behaviour [2]. The African continent accounts for only 4.8% of the world rice production, and despite its potential with 130 million hectares of arable soil, only 8% are currently under cultivation [3]. The average rice yield in Africa (2 Mg·ha$^{-1}$) is much lower than in Asian countries, e.g., Indonesia, Bangladesh, Vietnam and Myanmar which produce 7 Mg·ha$^{-1}$ [4].

In Mozambique, rice is cultivated over a total area of 320,000 hectares, thus representing the second most important source of cereal production. The average productivity, however, is low (about 1.04 Mg·ha$^{-1}$) [5], partly due to inappropriate or poor land preparation (burning crop residues, as well as the use of fire for opening new areas) and inadequate fertilizer management such as rate and timing of fertilizer application. This occurs mainly because most of this production is carried out by smallholder farmers who are engaged in rudimentary subsistence agriculture, depending mostly on rain as water source, use of local rice varieties, low technical support in management of soil fertility and minimal use of pesticides and fertilizers [6].

It is estimated that around 4.2 million small farms consist of subsistence-level rainfed agriculture and less than 1% of producers are commercial [5,7]. The majority of smallholders cannot be competitive by ensuring continuous supply in line with national rice consumption, so the country relies on imports to respond to the existing demand and low supply [8]. The lack of up-to-date and accessible information on profitable technology options for different producer groups in combination with poor provision of an effective agricultural extension service, have been pointed out as other potential factors affecting rice productivity in the country [9]. Inadequate fertilizer input is one of the most limiting factors to rice production. Furthermore, water is another vital factor that constrains rice production, namely water access, and additionally, weak water management capacity in the fields to guarantee timely adequate supply [10]. The fertilizer application rate in Mozambique has remained at less than 5.7 kg·ha$^{-1}$ [10], a cause for the consistently low production.

Intensive rice production and future rice demand will require intensive knowledge-based farming strategies for the efficient use of all inputs, including fertilizer nutrients. Increase in rice production requires an adequate amount of essential nutrients [11]. The use of integrated fertilization combining mineral and organic fertilizers has a strong potential and could be available and affordable for smallholder rice producers. An advantage of the application of organic wastes as fertilizer is that they usually provide nutritive elements to crops at little added cost along with the addition of organic matter to enrich the soil [12]. Compared with chemical fertilizers, nutrient availability is dependent on the mineralization rate of the organic material, which is generally low due to the high C/N ratio of these products leading to low crop growth and yield [13]. According to Iqbal et al. [14], organic fertilizers with lower nutrient releasing ability limit uptake of nutrients and fail to meet short-term crop requirements. Intensive rice production and future rice demand will require intensive knowledge-based farming strategies for the efficient use of all inputs, including fertilizer nutrients. Increase in rice production requires an adequate amount of essential nutrients [11]. The use of integrated fertilization combining mineral and organic fertilizers has a strong potential and could be available and affordable for smallholder rice producers. An advantage of the application of organic wastes as fertilizer is that they usually provide nutritive elements to crops at little added cost along with the addition of organic matter to enrich the soil [12]. Compared with chemical fertilizers, nutrient availability is dependent on the mineralization rate of the organic material, which is generally low due to the high C/N ratio of these products leading to low crop growth and yield [13]. According to Iqbal et al. [14] organic fertilizers with lower nutrient releasing ability limit uptake of nutrients and fail to meet short-term crop requirements.

Nitrogen is a macronutrient with an important role in rice production, compared to other nutrients [15], since it is fundamental to promote rapid plant growth and it improves grain yield and quality [16]. Therefore, better N management is essential, namely, the use of adequate (1) amounts of fertilizers, (2) N forms and formulations and (3) application time and method [17]. Nitrogen is a macronutrient with an important role in rice production, compared to other nutrients [15] since it is fundamental to promote rapid plant growth and it improves grain yield and quality [16]. Therefore, better N management is essential, namely the use of adequate (1) amounts of fertilizers, (2) N forms and formulations and (3) application time and method [17].

The continued increase in rice demand should be supported by increased production using adequate techniques, particularly in the developing countries [18]. Reducing dependence on imports, optimizing the use of existing resources in rice cultivation, with direct consequences for food security, generating wealth sources and economic growth of the Mozambican population are all a priority. Due to the high cost of chemical fertilizers in African countries, including Mozambique, it is necessary to guarantee an optimal and alternative way of rice production. However, few studies provide technical support related to crop productivity, and simultaneously, N use efficiency, which are important parameters to evaluate nutrient management in farms producing rice. Therefore, fertilization strategies will have to be developed and tested to increase rice production with low environmental impact and high acceptance from the producers. The main objective of the present study was to evaluate the agronomic effect and economic benefits of combined fertilization of rice production using mineral fertilizers and organic materials as N sources in Mozambican conditions.

## 2. Materials and Methods

### 2.1. Study Site

The study was conducted at the Chókwè Irrigation Scheme (CIS), located in Chókwè District, Gaza Province, adjacent to Limpopo River, southern Mozambique (24°52′ S, 33°00′ E, 33 m above sea level) [18]. The area is the major irrigation scheme in Mozambique [19] and covers 33,000 ha extension of land possible for irrigation [20]. The climate is predominantly semi-arid [21] with a mean annual temperature ranging from 22 to 26 °C and a total annual precipitation ranging from 500 to 700 mm. Yearly evapotranspiration ranges from 99.6 to 167.4 mm. The study was conducted at the Chókwè Irrigation Scheme (CIS), located in Chókwè District, Gaza Province, adjacent to Limpopo River, southern Mozambique (24°52′ S, 33°00′ E, 33 m above sea level) [18]. The area is the major irrigation scheme in Mozambique [19] and covers 33,000 ha extension of land possible for irrigation [20]. The climate is predominantly semi-arid [21] with a mean annual temperature ranging from 22 to 26 °C and a total annual precipitation ranging from 500 to 700 mm. Yearly evapotranspiration ranges from 99.6 to 167.4 mm.

Variation of precipitation, temperature and relative humidity measured during the experimental period (December–April) are presented in Table 1. Total rainfall was equivalent to 3371 mm. Precipitation mainly fell during January to February contributing 54% of the total rain during the experimental period. The average minimum and maximum temperatures are 20.0 °C (in April) and 33.4 °C (in March), respectively. The annual interval of relative humidity is 60.6 to 74.0%

**Table 1.** Average temperature, relative humidity and total rainfall registered during the field experiment (2018–2019).

| Year (2018) | Maximum Temperature (°C) | Minimum Temperature (°C) | Relative Humidity (%) | Rainfall (mm) |
|---|---|---|---|---|
| November | 32.7 | 22.1 | 62.5 | 370 |
| December | 32.8 | 28.2 | 60.8 | 741 |
| **Year (2019)** | | | | |
| January | 32.5 | 20.5 | 74.0 | 900 |
| February | 32.3 | 27.3 | 53.8 | 780 |
| March | 33.4 | 31.0 | 54.7 | 760 |
| April | 31.2 | 20.0 | 66.7 | 190 |
| May | 29.8 | 16.6 | 60.6 | 270 |

Rice is a primary crop cultivated during the rainy season (October to April) followed by fresh vegetables like tomatoes, onions, and green pepper cultivated during the dry season (May to September) [22]. Soil type was classified as loam sandy soil. The texture

of the top (0–20 cm) soil layer was classified as sandy loam (sand = 59%, silt = 30%, clay = 7%). Soil pH and total N concentrations in the top soil layer were 7.0 and 1.1 g.kg$^{-1}$, respectively. Cation exchange capacity and soil organic matter were 31.6 cmolc.kg$^{-1}$ and 21.9 g.kg$^{-1}$, respectively. Before the experiment, beef cattle manure and poultry litter samples were collected from the local farmers and analyzed to determine their physical and chemical properties. Detailed initial soil, beef cattle manure and poultry litter properties are presented in Table 2. Air temperature and precipitation data were collected from the weather station in the Chókwè meteorological office.

**Table 2.** Properties of the soil and organic fertilizer used in the experiment.

| Soil Parameters | Soil | Beef Cattle Manure | Poultry Litter |
|---|---|---|---|
| pH | 7 | 8.2 | 8.5 |
| Electrical conductivity (EC) ($\mu$S.cm$^{-1}$) | 640 | 7.2 | 4.4 |
| Dry matter (%) | NA | 84 | 80 |
| Total N (g.kg$^{-1}$) | 1.1 | 21 | 27.1 |
| NH$_4^+$-N (g.kg$^{-1}$) | 0.01 | 12 | 11 |
| NO$_3^-$-N (g.kg$^{-1}$) | 0.02 | 7.7 | 15 |
| CEC (cmolc.kg$^{-1}$) | 31.63 | 30.2 | 32.1 |
| C/N ratio | 11.55 | 12 | 10 |
| Organic matter (g.kg$^{-1}$) | 21.9 | 31.8 | 30.7 |
| P (mg.kg$^{-1}$) | 33.69 | 12.1 | 12.3 |
| K (mg.kg$^{-1}$) | 660 | NA | NA |
| Exchangeable cations (cmol.kg$^{-1}$) | | | |
| Ca | 17.5 | NA | NA |
| Mg | 9.94 | NA | NA |
| K | 1.26 | NA | NA |
| Na | 1.88 | NA | NA |

(Mean of three replicates; values presented on a dry matter basis); NA- not available.

### 2.2. Treatments and Experimental Field Design

From December 2018 to April 2019, seven treatments (four replicates) were set up in terms of N content: T0: no fertilizer, T1: 100% urea, T2: 100% beef cattle manure, T3: 100% poultry litter, T4: 50% urea + 50% beef cattle manure, T5: 50% urea + 50% poultry litter and T6: 40% urea + 30% beef cattle manure + 30% poultry litter. Details of each treatment are presented in Table 3. Twenty-eight plots of 2m × 3m were used for the present experiment. All plots were separated by 50 cm ridges to avoid the mixing of water and nutrients from adjacent treatments. All treatments were managed following the same practices (irrigation, weeding pest and disease control) usually adopted by farmers during the rice growing season. Different amounts of fertilizers were applied to supply a total amount of N, equivalent to 100 kg·ha$^{-1}$, except T0, which received no fertilizer. Fertilizers were applied in two distinct time periods: half was applied two weeks before transplanting (applied as basal fertilization) and the remaining/other half was applied before panicle maturity stage. Irrigation water was separately applied to each treatment from a water channel. The irrigation requirements were conducted based on rice plants needs and weather conditions. Rice seeds (cv Makassane) were sown in a nursery on 18 December 2018 and seedlings were transplanted on 10 January 2019.

**Table 3.** Amounts of natural and synthetic organic N-fertilizers applied in each treatment.

| Treatment | Amount of Fertilizer Applied (kg·ha$^{-1}$) | | | Total N Applied (kg·ha$^{-1}$) |
|---|---|---|---|---|
| | Urea | Beef Cattle Manure | Poultry Litter | |
| T0 | 0 | 0 | 0 | 0 |
| T1 | 1.3 | 0 | 0 | 100 |
| T2 | 0 | 3.0 | 0 | 100 |
| T3 | 0 | 0 | 2.2 | 100 |
| T4 | 0.7 | 1.5 | 0 | 100 |
| T5 | 0.7 | 0 | 1.1 | 100 |
| T6 | 0.5 | 9 | 6.6 | 100 |

*2.3. Soil Analysis*

Soil samples were collected from the 0–20 cm topsoil layer at harvesting time. Soil pH and electrical conductivity (EC) were determined in a soil:water (1:2.5 w/v) suspension. Soil samples were extracted with 2M potassium chloride (KCl) solution and the extract was analyzed for ammonium N ($NH_4^+$-N) and nitrate N ($NO_3$-N) concentration by continuous flow analysis as described by [23]. The N total was measured by Kjeldahl method [24]. The organic matter was quantified by loss-on-ignition after incineration at 500–550 °C of dry sample. Potassium (K) and phosphorus (P) content were determined according to Egner–Riehm method [25]. Available Cu, Zn, Fe and Mn were extracted based on the procedure described by Lakanen and Ervio (1971) [26] and quantified by atomic absorption spectrophotometer. Exchangeable cations were extracted by 1M ammonium acetate at pH 7 and measured by atomic absorption spectrophotometry [27].Soil samples were collected from the 0–20 cm topsoil layer at harvesting time. Soil pH and electrical conductivity (EC) were determined in a soil:water (1:2.5 w/v) suspension. Soil samples were extracted with 2M potassium chloride (KCl) solution and the extract was analysed for ammonium N ($NH_4^+$-N) and nitrate N ($NO_3$-N) concentration by continuous flow analysis as described by [23]. The N total was measured by Kjeldahl method [24]. The organic matter was quantified by loss-on-ignition after incineration at 500–550 °C of dry sample. Potassium (K) and phosphorus (P) content were determined according to Egner-Riehm method [25]. Available Cu, Zn, Fe and Mn were extracted based on the procedure described by Lakanen and Ervio (1971) [26] and quantified by atomic absorption spectrophotometer. Exchangeable cations were extracted by 1M ammonium acetate at pH 7 and measured by atomic absorption spectrophotometry [27].

*2.4. Growth Indicators, Plant Yield and Nutrient Uptake*

From the day of transplanting up to harvesting time on 17 May 2019, plant growth parameters such as plant height, and number of panicles in the same plants, were collected every two weeks throughout the experiment. Ten plants were randomly selected in each plot and identified to allow the monitoring of these two growth parameters. At harvest time, the plant height, the thousand-grain weight (TGW), the number of spikelets per panicle and the maximum panicle length (cm) were measured in these same ten plants.

At the end of the experiment, aboveground (straw and grain) and belowground (roots) samples were harvested manually in an area of 1 m$^2$ (1 m × 1 m) randomly selected in each plot. In this sampling area, panicles of each plant were counted, and all samples were air dried for almost one week. Grains were separated from straw and the total number of grains was counted and weighed. Rice straw was also weighed to evaluate the total dry matter (grain + straw).

Subsamples of each plant component (grain, straw and roots) were ground and analyzed for N content by micro-Kjeldahl method [24] and for K and P content by hydrochloric acid digestion followed by quantification by the ammonium vanadomolybdate method for P and molecular absorption spectrophotometry for K [28]. Crop nutrient uptake was calculated from dry biomass weight and percentage of nutrient content (grain and straw) [29].Subsamples of each plant component (grain, straw and roots) were ground

and analysed for N content by micro-Kjeldahl method [24] and for K and P content by hydrochloric acid digestion followed by quantification by the ammonium vanadomolybdate method for P and molecular absorption spectrophotometry for K [28]. Crop nutrient uptake was calculated from dry biomass weight and percentage of nutrient content (grain and straw) [29].

### 2.5. Statistical Analysis

Data were subjected to analysis of variance (ANOVA) [30] and means between treatments were compared using the Tukey's honest significance difference test (HSD) at 5% level of significance. The analyses were carried out using "R" Software (version 3.3.2).

## 3. Results

### 3.1. Plant Growth Parameters

Plant growth parameters are important tools to assess rice productivity and yield. At harvest time, the main parameters related to rice growth assessed here were plant height, tillers per hill, number of spikelets per panicle, and the yield components. The number of tillers also indicate the number of panicles, so these parameters give one idea about yield potential. Likewise, number of spikelets per panicle is one tool used to estimate rice yield in experiments. Some parameters such as number of tillers per hill and rice plant height were assessed also across the experiments. This information allows to estimate the performance that the plant can have to achieve yield attributes since excellent vegetative growth and development resulted in maximum height.

Variation of plant height and number of tillers during the experiment are presented in Figures 1 and 2, respectively. Plant height throughout the experiment followed a similar trend: T1 > T6 > T5 > T4 > T3 > T2 > T0 (Figure 1), with no significant differences. Maximum plant height was observed in T1 and T6, at 114.5 and 113.25 cm, respectively, whereas minimum plant height was observed in T0 at 100.25 cm. Similar plant heights were observed by Moe et al. [13] where rice was fertilized with urea alone and manure plus urea.

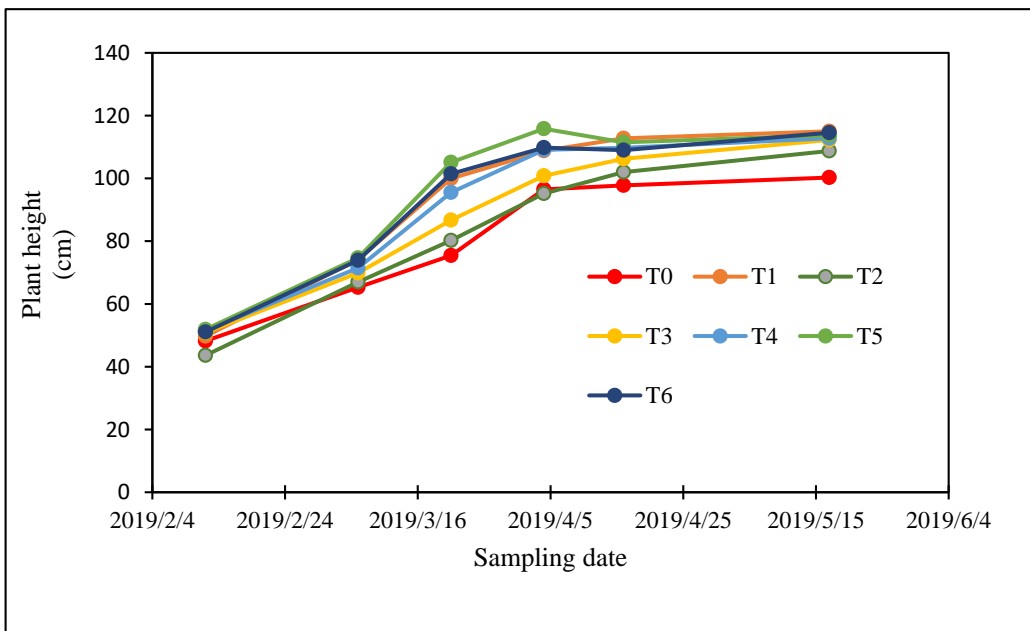

**Figure 1.** Effect of the different treatments on rice plant height; mean of 4 replicates; error bars were removed for clarity, T0: no fertilizer, T1: 100% urea, T2: 100% beef cattle manure, T3: 100% poultry litter, T4: 50% urea + 50% beef cattle manure, T5: 50% urea + 50% poultry litter and T6: 40% urea + 30% beef cattle manure + 30% poultry litter.

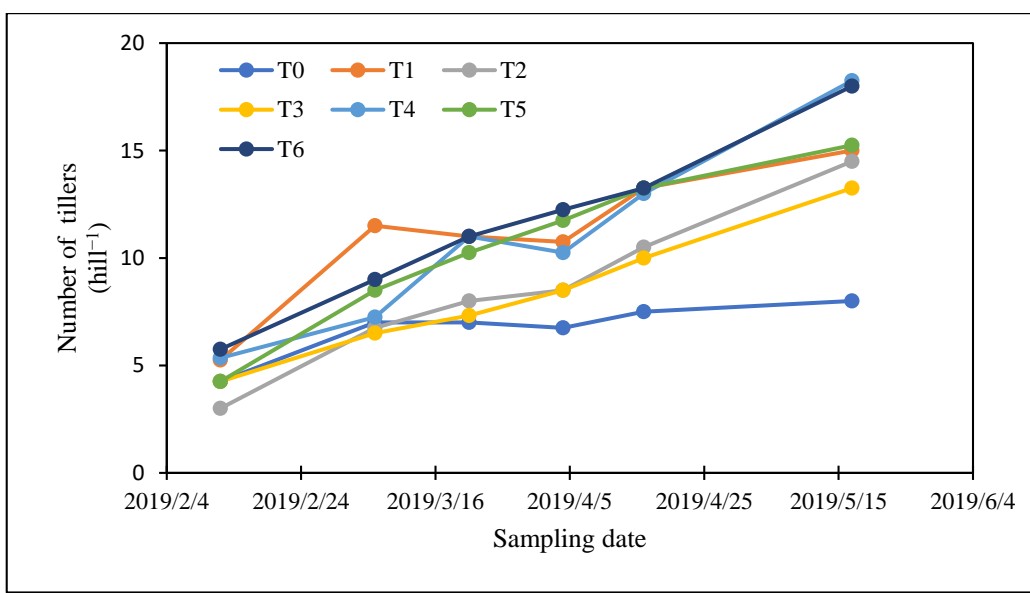

**Figure 2.** Effect of the different treatments on rice plant tillering; mean of 4 replicates; error bars were removed for clarity, T0: no fertilizer, T1: 100% urea, T2: 100% beef cattle manure, T3: 100% poultry litter, T4: 50% urea + 50% beef cattle manure, T5: 50% urea + 50% poultry litter and T6: 40% urea + 30% beef cattle manure + 30% poultry litter.

Maximum tillering was observed in T6 and T4, with 18 and 16 tillers, respectively. Minimum tillering was observed in T0 with 8 tillers (Figure 2). The values obtained here were similar to those obtained by John Hunter et al. [31].

The 1000-grain weight was also affected by the treatments (Figure 3) and was significantly ($p < 0.05$) greater in all the fertilized treatments compared to the control. The highest 1000-grain weight was obtained in T2 and T6 with a significantly higher value compared to the remaining treatments. The lowest values in fertilized treatments were observed in T3 and T4.

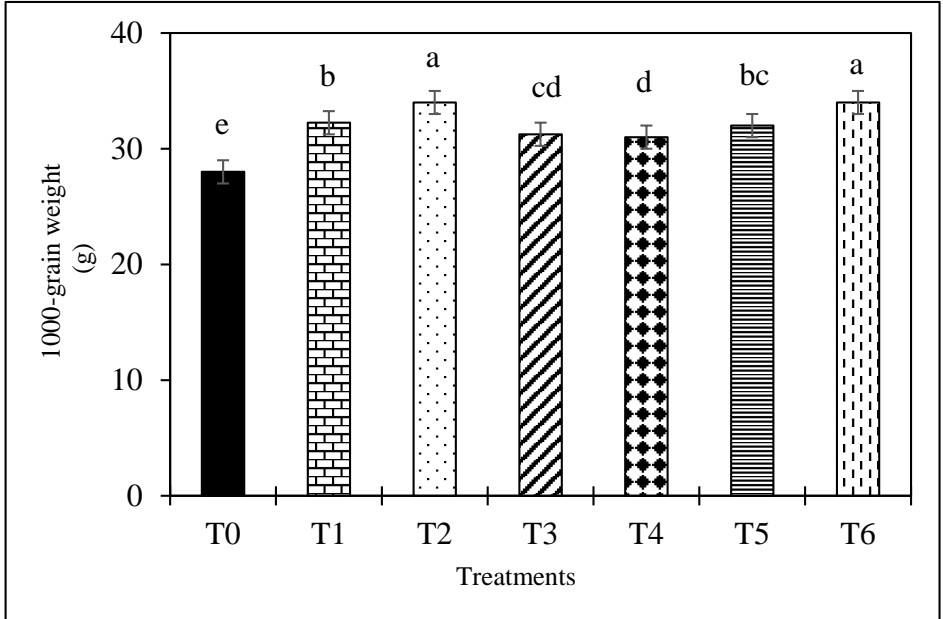

**Figure 3.** Effect of different N fertilizer and their combinations on one thousand grain weight; mean of 4 replicates. Different letters indicate statistical differences ($p < 0.05$), T0: no fertilizer, T1: 100% urea, T2: 100% beef cattle manure, T3: 100% poultry litter, T4: 50% urea + 50% beef cattle manure, T5: 50% urea + 50% poultry litter and T6: 40% urea + 30% beef cattle manure + 30% poultry litter.

The number of spikelets was significantly influenced by the different N sources among treatments (Figure 4). The highest number of spikelets (96) was observed in T6, even if no significant differences were observed between T6, T5, T4 and T3. Significant differences were observed between T0, T1 and T2 where the lowest number of spikelets was observed (72) (Figure 4). A high number of spikelets observed in the integrated treatments indicates that the combined application of fertilizers caused increased growth. This may have been due to enhanced mineralization of the organic N from the manure indicating that the combination with chemical fertilizers is efficient [32].

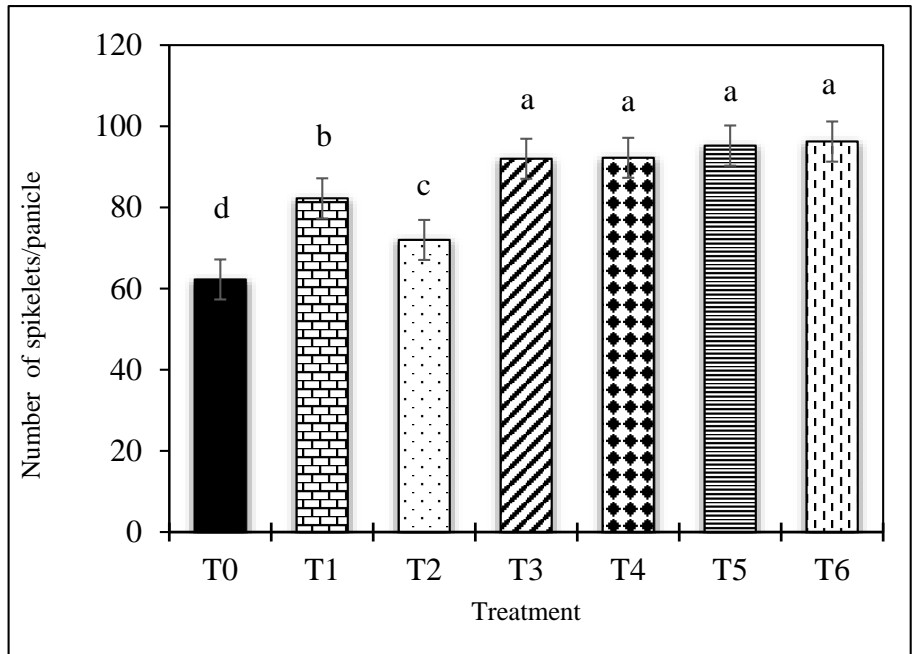

**Figure 4.** Effect of the different treatment combinations of N source on number of spikelets per panicle; mean of 4 replicates. Different letters indicate statistical differences ($p < 0.05$), T0: no fertilizer, T1: 100% urea, T2: 100% beef cattle manure, T3: 100% poultry litter, T4: 50% urea + 50% beef cattle manure, T5: 50% urea + 50% poultry litter and T6: 40% urea + 30% beef cattle manure + 30% poultry litter.

### 3.2. Rice Yield and Yield Components

Our results demonstrate that grain yield was significantly affected by fertilizer application compared to plants which received no fertilizer. However, there was no significant statistical difference between T1, T4, T5 and T6. The highest grain yields were observed in T6, followed by T5 > T4 > T1 > T2 > T3 > T0. The lowest grain yield of 200 g·m$^{-2}$ was recorded in plants treated with no fertilizer while a yield of 425 g·m$^{-2}$ was observed in T6 (Figure 5).

An overview of the effects of treatments T2 to T6 on rice experimental parameters compared with urea alone (T1) is given in (Table 4). As can be seen, some parameters such as number of tillers and spikelets and yield were positively affected by the integrated treatment compared to urea alone. Application of integrated fertilizers (T4 and T6) increased the number of tillers by 20%. There was no increase in plant height in any treatment compared with T1. The sole use of poultry litter (T3) and combined use of urea and manure fertilizers (T4, T5 and T6) increased the number of spikelets by 12, 12, 16 and 17% relative to T1, respectively. Application of beef cattle manure (T2) and integrated fertilization (T6) showed an increase of 6% in 1000-grain weight. Application of integrated fertilizers (T4, T5 and T6) increased yield by 3, 6, and 11%, respectively. Plants treated with T4, T5 and T6 also had a positive response in numbers of tillers and spikelets and yield compared with the plants treat with only urea.

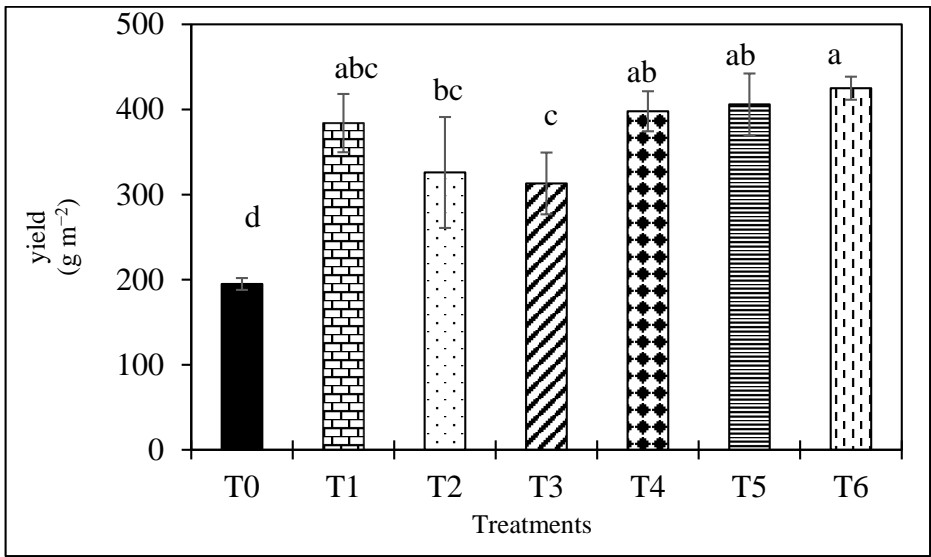

**Figure 5.** The effect of different N fertilizers and their combinations on rice grain yield; mean of 4 replicates. Different letters indicate statistical differences ($p < 0.05$), T0: no fertilizer, T1: 100% urea, T2: 100% beef cattle manure, T3: 100% poultry litter, T4: 50% urea + 50% beef cattle manure, T5: 50% urea + 50% poultry litter and T6: 40% urea + 30% beef cattle manure + 30% poultry litter.

**Table 4.** Effect of applied treatments (T2: 100% beef cattle manure, T3: 100% poultry litter, T4: 50% urea + 50% beef cattle manure, T5: 50% urea + 50% poultry litter and T6: 40% urea + 30% beef cattle manure + 30% poultry litter) on different parameters compared with T1 (100% urea).

| Treatments | Plant Height | Number of Tillers | Number of Spikelets | 1000-Grain Weight | Yield |
|---|---|---|---|---|---|
| T2 | ⇒ | ⇒ | ⇒ | ⇗ | ⇒ |
| T3 | ⇒ | ⇒ | ⇗ | ⇒ | ⇒ |
| T4 | ⇒ | ⇗ | ⇗ | ⇒ | ⇗ |
| T5 | ⇒ | ⇗ | ⇗ | ⇒ | ⇗ |
| T6 | ⇒ | ⇗ | ⇗ | ⇗ | ⇗ |

⇒ : no effect; ⇗ : an increase.

## 3.3. Nutrient Content

The effects of the six treatments on the nutrient content of the rice plants (root, grain and straw samples) are shown in (Table 5). For root nutrient content, highest level of root Mg was observed in T5 at 3956.09 mg.kg$^{-1}$ which was not statistically different from all the treatments. The lowest amount of Mg content in root was found in the control (T0).

The treatments T0 to T3 produced levels of Fe in roots that were up to 7-fold higher than in treatments T4 to T6.

For grain nutrient content, although there were no significant differences between the treatments, highest levels of grain N were observed in T5 and T6, while the lowest was observed in T1. P accumulation was lowest in T5 and highest in T1. K accumulation was highest in T6 and lowest in T2.

**Table 5.** Effects of fertilization treatments on nutrient accumulation in the rice plants; mean of 4 replicates.

| Treatments | Mineral Content in Rice Straw (mg.kg$^{-1}$) | | | | | | | | | |
|---|---|---|---|---|---|---|---|---|---|---|
| | N | P | K | Ca | Mg | Na | Fe | Cu | Zn | Mn |
| T0 | 7204.25 c | 1557.66 b | 5125.13 | 5709.3 | 1189.59 | 7204.25 c | 2378.26 | 42.32 | 55.4 | 817.43 c |
| T1 | 9148.58 ab | 1895.21 ab | 5095.28 | 6238.14 | 2661.3 | 9148.59 ab | 3734.54 | 54.33 | 57.94 | 837.49 b |
| T2 | 8855.76 abc | 2016.21 a | 5309.3 | 5955.49 | 199.24 | 8855.77 abc | 3137.8 | 54.74 | 56.85 | 773.49 e |
| T3 | 8881.07 abc | 1949.49 ab | 5486.29 | 6552.12 | 4981.6 | 8881.08 abc | 3129.38 | 55.64 | 56.66 | 790.34 d |
| T4 | 8052.95 bc | 1804.18 ab | 5433.73 | 6349.82 | 232.15 | 8052.95 bc | 2323.57 | 54.74 | 62.27 | 870.35 a |
| T5 | 9559.71 ab | 1917.69 ab | 5515.91 | 6436.95 | 2578.82 | 9559.71 ab | 3414.07 | 55.24 | 58.09 | 775.51 de |
| T6 | 10,155.7 a | 1887.56 ab | 5146.55 | 6204.23 | 2197.62 | 10,155.79 a | 2282.46 | 54.14 | 56.66 | 822.55 bc |

| Treatments | Mineral content in rice grain (mg.kg$^{-1}$) | | | | | | | | | |
|---|---|---|---|---|---|---|---|---|---|---|
| | N | P | K | Ca | Mg | Na | Fe | Cu | Zn | Mn |
| T0 | 8569.73 | 20.54 | 5.25 | 1.26 | 8.42 | 0.61 a | 0.31 | 0.18 | 2.24 | 0.25 |
| T1 | 8542.61 | 24.78 | 5.52 | 1.26 | 10.39 | 0.34 a | 0.45 | 0.16 | 2.2 | 0.29 |
| T2 | 9279.55 | 21.2 | 5.16 | 1.16 | 8.61 | 0.29 a | 0.4 | 0.14 | 2.37 | 0.24 |
| T3 | 9005.85 | 22.81 | 5.59 | 1.24 | 9.23 | 0.42 a | 0.25 | 0.16 | 2.33 | 0.26 |
| T4 | 8991.96 | 21.38 | 5.27 | 1.13 | 8.16 | 0.29 a | 0.39 | 0.14 | 2.28 | 0.26 |
| T5 | 9532.9 | 19.55 | 5.04 | 1.08 | 7.23 | 0.04 a | 0.37 | 0.15 | 2.34 | 0.21 |
| T6 | 9354.42 | 20.94 | 5.29 | 1.15 | 8.06 | 0.17 a | 0.31 | 0.13 | 2.37 | 0.25 |

| Treatments | Mineral content in rice root (mg.kg$^{-1}$) | | | | | | | | | |
|---|---|---|---|---|---|---|---|---|---|---|
| | N | P | K | Ca | Mg | Na | Fe | Cu | Zn | Mn |
| T0 | 5093.6 | 5412.94 | 6878.22 | 3952.77 | 2773.8 b | 3400.83 | 20,781.45 a | 40.99 | 55.57 | 509.82 |
| T1 | 3589.01 | 5936.93 | 8919.15 | 3792.8 | 3258.7 ab | 4292.52 | 20,310.99 a | 58.02 | 54.63 | 545.29 |
| T2 | 4326.9 | 5578.15 | 8242.13 | 3890.35 | 2860.8 ab | 3881.94 | 23,101.07 a | 34.76 | 96.04 | 525.37 |
| T3 | 5060.25 | 5496.22 | 8012.77 | 4077.31 | 3393.85 ab | 3387.57 | 19,275.17 a | 46.48 | 58.42 | 567.86 |
| T4 | 4423.99 | 5758.9 | 6982.94 | 4244.45 | 3322.39 ab | 3813.65 | 7429.72 b | 44.41 | 75.04 | 525.56 |
| T5 | 4867.76 | 6183.64 | 6939.77 | 4422.64 | 3956.09 ab | 2735.8 | 3095.58 b | 38.15 | 47.46 | 686.52 |
| T6 | 5409.26 | 6161.38 | 8182.94 | 3726.39 | 3291.11 ab | 3428.45 | 3530.92 b | 31.72 | 86.24 | 590.05 |

Means followed by the same letter in a column do not differ from each treatment by Tuckey´s test at ($p < 0.05$); T0: no fertilizer, T1: 100% urea, T2: 100% beef cattle manure, T3: 100% poultry litter, T4: 50% urea + 50% beef cattle manure, T5: 50% urea + 50% poultry litter and T6: 40% urea + 30% beef cattle manure + 30% poultry litter.

As shown in (Table 5, nutrient contents in straw were significantly influenced by the treatments for some macro and micronutrients, namely, N, P, Na and Mn. The highest amount of N in straw was observed in T6 at 10,155.7 mg.kg$^{-1}$ which was statistically different from all other treatments. As expected, the lowest amount of N content in straw was observed in the control (T0). With respect to P concentration in straw, the values ranged from 1557 mg.kg$^{-1}$ in T0 to 2016 mg.kg$^{-1}$ in T2, statistically different from all other treatments. Additionally, the next highest P level was observed in T3 at 1949 mg.kg$^{-1}$ which was statistically similar to all other treatments. Considering Na in straw, the highest level was observed in T6 which was statistically different from all other treatments. The next highest result was observed in T5.

Finally, the highest amount of Mn in straw was observed in T4 which was statistically different from all the other treatments while the lowest result was observed in T2 at 773 mg.kg$^{-1}$.

### 3.4. Nutrient Uptake by Rice

Total nutrient uptake (N, P and K) by the rice plants as well as specific uptake in root, grain and straw are detailed in Table 6. Our data indicate that there are significant differences ($p < 0.05$) in the total N uptake ranging from 31.56 to 68.87 kg·ha$^{-1}$. Highest total N uptake was observed in T5 and T6 which was statistically different from the other treatments. Lowest total N uptake was observed in T0, the control, as expected.

**Table 6.** Effect of fertilizer on N, P and K uptake (kg·ha$^{-1}$) in rice plants.

| Treatment | N uptake (kg·ha$^{-1}$) | | | Total |
| --- | --- | --- | --- | --- |
| | Root | Grain | Straw | |
| T0 | 5.23 a | 17.58 b | 8.75 d | 31.56 c |
| T1 | 6.98 a | 28.94 a | 17.64 ab | 53.56 ab |
| T2 | 6.59 a | 29.47 a | 13.07 bcd | 49.12 b |
| T3 | 6.97 a | 27.71 ab | 12.15 cd | 46.83 b |
| T4 | 8.50 a | 34.79 a | 15.44 abc | 58.73 ab |
| T5 | 9.85 a | 38.08 a | 18.93 a | 66.87 a |
| T6 | 10.16 a | 37.67 a | 18.98 a | 66.81 a |

| Treatment | P uptake (kg·ha$^{-1}$) | | | Total |
| --- | --- | --- | --- | --- |
| | Root | Grain | Straw | |
| T0 | 6.55 c | 2.54 c | 1.89 b | 10.99 c |
| T1 | 11.52 ab | 4.80 a | 3.67 a | 19.98 a |
| T2 | 8.07 bc | 2.98 bc | 2.96 ab | 14.01 bc |
| T3 | 7.54 bc | 3.13 bc | 2.68 ab | 13.35 bc |
| T4 | 11.01 ab | 4.07 ab | 3.43 a | 18.52 ab |
| T5 | 12.47 a | 3.90 abc | 3.90 a | 20.27 a |
| T6 | 11.41 ab | 3.82 abc | 3.54 a | 18.77 ab |

| Treatment | K uptake (kg·ha$^{-1}$) | | | Total |
| --- | --- | --- | --- | --- |
| | Root | Grain | Straw | |
| T0 | 8.30 b | 5.66 b | 6.23 c | 20.19 b |
| T1 | 17.29 a | 10.22 ab | 9.86 abc | 37.36 a |
| T2 | 12.23 ab | 8.43 ab | 7.86 abc | 28.52 ab |
| T3 | 11.01 ab | 9.63 ab | 7.52 bc | 28.16 ab |
| T4 | 13.33 ab | 10.75 a | 10.38 ab | 34.46 a |
| T5 | 14.39 ab | 10.15 ab | 11.21 a | 35.75 a |
| T6 | 15.22 ab | 11.43 a | 9.65 abc | 36.30 a |

Means followed by the same letter in a column do not differ from each other using Tuckey´s test at ($p < 0.05$) significance. T0: no fertilizer, T1: 100% urea, T2: 100% beef cattle manure, T3: 100% poultry litter, T4: 50% urea + 50% beef cattle manure, T5: 50% urea + 50% poultry litter and T6: 40% urea + 30% beef cattle manure + 30% poultry litter.

Regarding the specific components of the rice plant, results showed a wide variation in N uptake by straw and grain but with the same trend observed in total N uptake. Regarding the root, higher total N uptake was observed in T5 and T6, although not statistically significant.

Total P uptake ranged from 10.99 to 20.27 kg·ha$^{-1}$ over the treatments with the lowest level in T0 (control). The average total P uptake was 20.27 kg·ha$^{-1}$ for T5 which was similar to 19.9 kg·ha$^{-1}$ for T1. Phosphorus uptake by roots ranged from 6.55 to 12.47 kg·ha$^{-1}$ with the highest level in T5. The lowest P uptake in root (6.55 kg·ha$^{-1}$) was observed in T0 (control). P uptake in grain ranged from 2.54 to 4.80 kg·ha$^{-1}$. Highest was found in T1 even if not statistically different from T4, T5 and T6. Regarding P uptake in straw, values varied from 1.89 to 3.90 kg·ha$^{-1}$ with the highest level observed in T5.

The highest values of total K uptake by the rice plants were achieved with combined fertilizers (17.28 kg·ha$^{-1}$ which was statistically similar to T4, T5 and T6). Lowest mean value was 8.30 kg·ha$^{-1}$ in the control. K uptake in root ranged from 8.30 to 17.29 kg·ha$^{-1}$ with the highest value in T1 (not statistically different from the other fertilized treatments). K uptake in grain ranged from 5.56 to 11.75 kg·ha$^{-1}$. Regarding K uptake in straw, values ranged from 6.23 to 11.21 kg·ha$^{-1}$.

### 3.5. Soil Properties after Harvest

Significant differences were observed between treatments in terms of N-NH$_4$$^+$, available P and available K while pH, EC, N-NO$_3$$^-$, OM, CEC presented similar values in all treatments at the end of the experiment (Table 7). The maximum N-NH$_4$$^+$ content of 9.9

mg.kg$^{-1}$ was obtained in the treatment receiving three sources of N (T6), followed by T2 and T3 with 8.9 and 8.8 mg.kg$^{-1}$, respectively.

**Table 7.** Properties of the soil after harvesting in the experiment.

| Treatment | pH | EC | N-NH$_4$ | N-NO$_3$ | P Available | K Available | OM | CEC | Na | K | Ca | Mg |
|---|---|---|---|---|---|---|---|---|---|---|---|---|
| | | (µS.cm$^{-1}$) | | | (mg.kg$^{-1}$) | | (mg.kg$^{-1}$) | cmol(+)/kg | | | cmol(+)/kg | |
| T0 | 7.4 | 620 | 7.3 e | 1.52 | 21.1e | 24.6 c | 2.6 | 36.3 | 1.09 | 0.47 | 2.43 b | 0.39 |
| T1 | 7.5 | 640 | 9.6 ab | 0.76 | 23.4d | 26.5 ab | 2.7 | 36.9 | 0.79 | 0.23 | 1.89 b | 0.45 |
| T2 | 7.6 | 620 | 8.8 bc | 1.16 | 21.1e | 25.8 b | 2.8 | 36.3 | 0.69 | 0.22 | 1.60 b | 0.37 |
| T3 | 7.7 | 640 | 8.9 bc | 1.49 | 26.1c | 24.8 c | 2.6 | 35.8 | 0.54 | 0.24 | 2.37 b | 0.47 |
| T4 | 7.8 | 620 | 7.5 de | 1.08 | 23.7d | 24.1 c | 2.8 | 35.8 | 0.71 | 0.39 | 4.51 a | 0.89 |
| T5 | 7.6 | 660 | 8.4 cd | 1.45 | 33.7b | 24.4 c | 2.8 | 37.6 | 0.69 | 0.31 | 4.68 a | 0.93 |
| T6 | 7.7 | 650 | 9.9 a | 0.89 | 35.7a | 27.2 a | 2.8 | 39.1 | 0.61 | 0.45 | 3.93 a | 0.91 |

Means followed by the same letter in a column do not differ from each treatment using Tuckey´s test at ($p < 0.05$) significance. T0: no fertilizer, T1: 100% urea, T2: 100% beef cattle manure, T3: 100% poultry litter, T4: 50% urea + 50% beef cattle manure, T5: 50% urea + 50% poultry litter and T6: 40% urea + 30% beef cattle manure + 30% poultry litter.

Organic matter content showed no significant differences between the treatments, despite the application of combined N sources which should favour a long-term increase in soil organic matter. Application of different N sources had no significant impact on the exchangeable cations (Mg, Na and K) even if Ca was significantly higher in T4, T5 and T6. pH tended to increase in all treatments even if no significant differences were observed. The combined application of manures (beef cattle and poultry litter) plus urea tended to increase the available P content compared with the control. Similarly, the highest content of available K was observed in T6. It can be concluded that a combination of several N sources tended to improve soil chemical properties.

*3.6. Economic Evaluation*

Table 8 presents the estimated costs (per hectare) of the implementation of the treatments in our study. Considering that both poultry and cattle manure are not marketed in the studied region, the amount that would be spent on transport to cultivation areas was considered for financial evaluation purposes, according to the data obtained by the authors during the study period. The unit price transport costs were 0.002 €.kg$^{-1}$ for poultry manure and 0.001 €.kg$^{-1}$ for cattle manure. The unit cost of synthetic urea was 0.58 €.kg$^{-1}$ [5]. To calculate the monetary yield for each treatment, the values of the rice weight per hectare were used. These values were then multiplied by the sale price (0.20 €.kg$^{-1}$). The difference between monetary income and the cost of production was considered profit.

**Table 8.** Economic evaluation.

| Treatment | Cost of Fertilizers Application (€·ha$^{-1}$) | Cost per Ton of Rice Produced | Monetary Income (€·ha$^{-1}$) | Profit (€·ha$^{-1}$) |
|---|---|---|---|---|
| T0 | 0 | 0 | 0 | 0 |
| T1 | 125.86 | 37.6 | 768 | 642.14 |
| T2 | 5 | 1.59 | 650 | 645 |
| T3 | 7.4 | 2.41 | 626 | 618.6 |
| T4 | 65.43 | 16.89 | 794 | 728.57 |
| T5 | 66.63 | 16.66 | 810 | 743.37 |
| T6 | 54.06 | 13.43 | 850 | 795.94 |

Other production costs were not included e.g., labour, seed price, ploughing land preparation, irrigation water tax, scaring of birds, weed control and herbicide application. These costs were similar across all the treatments and independent of the type of fertilizer applied.

The highest cost per hectare was for T1 (urea only) at 125.86 €·ha$^{-1}$. The results showed that the similar yields found in T5 and T6 had different costs. Cost for T5 (66.63 €·ha$^{-1}$) was higher than that for T6 (54.06 €·ha$^{-1}$), indicating that combining cattle, poultry

and urea was more profitable. The use of combined fertilizers (T4, T5 and T6) produced higher income than the use of sole fertilizers (T1, T2 and T6).

The highest profit of 795.94 €·ha$^{-1}$ was obtained in the treatment receiving three sources of N, followed by T5 and T4 with 743.37 €·ha$^{-1}$ and 728.57 €·ha$^{-1}$, respectively.

## 4. Discussion

### 4.1. Effect of the Fertilizer Treatments on Rice Growth Parameters

Our results indicate that single applications of urea or manure as well as their combined application had a positive influence on the vegetative growth of the crop. Maximum plant heights were observed in T1, T6, T5 and T4 ($p < 0.05$), followed by T3 and T2. Similar tendencies were observed in number of tillers, with higher numbers in T4 and T6 (18 tillers) and lower numbers in T2 and T3 (14 and 13 tillers, respectively). These results concur with Moe et al. [33], who reported that combined fertilizers had a significant positive effect on height, spikelets per sq m, tillering and grain yield. Karki et al. [34] emphasized that a combined application of cattle manure with chemical fertilizer and even poultry manure, is an effective approach to enhance rice growth. Our results indicate that single applications of urea or manure as well as their combined application had a positive influence on the vegetative growth of the crop. Maximum plant heights were observed in T1, T6, T5 and T4 ($p < 0.05$), followed by T3 and T2. Similar tendencies were observed in number of tillers, with higher numbers in T4 and T6 (18 tillers) and lower numbers in T2 and T3 (14 and 13 tillers, resp.). These results concur with Moe et al. [33] who reported that combined fertilizers had a significant positive effect on height, spikelets per sq m, tillering and grain yield. Karki et al. [34] emphasized that a combined application of cattle manure with chemical fertilizer and even poultry manure, is an effective approach to enhance rice growth.

Thousand-grain weight was significantly higher ($p < 0.05$) relative to the control in treatments fertilized with the three N sources as well as in the treatment with only cattle manure. The values achieved here were higher than those obtained by Moe et al. and Sing et al. [13,32]. Similar findings were observed by Ghoneim et al. [35], who report that N fertilizer application significantly increased the 1000-grain weight, due to the production of a higher number of spikelets per panicle in the plants. Thousand-grain weight was significantly higher ($p < 0.05$) relative to the control in treatments fertilized with the three N sources as well as in the treatment with only cattle manure. The values achieved here were higher than those obtained by Moe et al. and Sing et al. [13,35]. Similar findings were observed by Ghoneim et al. [36] which reports that N fertilizer application significantly increased the 1000-grain weight, due to the production of a higher number of spikelets per panicle in the plants.

The grain yield collected from the field: experiments showed a variable response to different fertilizer treatments (Figure 5), T5 and T6 produced, indeed, significantly ($p < 0.05$) greater amounts of grain compared with the control treatment. There was no significant statistical difference between T1, T4, T5 and T6. The higher grain yield was observed in treatments with combined sources of N in agreement with Amanullah et al. [36], who reported the combined application of manure and urea N was better than a single organic source to increase grain yield and yield components. This fact can also be explained by the use of mineral fertilizers combined with manure, where organic N mineralization kept nutrient stress arrested through the entire plant growing period resulting in higher grain production [37]. The grain yield collected from the field; experiments showed a variable response to different fertilizer treatments (Figure 5), T5 and T6 produced indeed significantly ($p < 0.05$) greater amounts of grain compared with the control treatment. There was no significant statistical difference between T1, T4, T5 and T6. The higher grain yield was observed in treatments with combined sources of N in agreement with Amanullah et al. [37] who reported the combined application of manure and urea N was better than a single organic source to increase grain yield and yield components. This fact can also be explained by the use of mineral fertilizers combined with manure where

organic N mineralization kept nutrient stress arrested through the entire plant growing period resulting in higher grain production [33].

Likewise, our results are further supported by Banik et al. [38] who stated that combined use of organic and synthetic fertilizers offers better synchrony of nutrient availability to the rice plant crop leading to higher biomass production and also to better nutrient use efficiency. A previous study conducted by Liu et al. [39] also highlighted that the combined application of manure and conventional N fertilizers could lead to a better yield than the use of solely conventional fertilizers.

One of the most limiting factors for improving production in tropical regions is related to the fast decomposition of organic matter with the negative consequence of poor nutrient retention [40]. Adding different sources of organic manure could therefore contribute to achieve higher yields by keeping nutrients in the soil for a longer period. Manure only or combined application of manure with urea could improve physicochemical soil proprieties, namely, soil fertility, soil porosity and water holding capacity [41]. In the present study, a positive impact on soil chemical proprieties was observed in terms of soil available P, available K, N-$NH_4^+$ and Ca content. Our results are in accordance with Liu et al. [39] who report that the decomposition of manure slowly releases nutrients to the soil and improved soil chemical properties.

The increase in soil Ca from initially 2.43 to 4.68 cmol(+)/kg following treatment with cattle and poultry manure + urea may be due to the addition of organic manure that influences cation exchange capacity [42].

Our results showed an increase of available P and N-$NH_4^+$ in the treatments with integrated fertilization. These findings are in concordance with Manitoba [43], who reports that application of solid manure moves soil pH towards neutrality in acidic and alkaline soils, thus improving availability of macronutrients like K and micronutrients.

Smallholder farmers may become resistant to new technologies, mainly due to economic reasons, hence the reason for our approach to adopt the treatments tested in this study. Our findings show less cost in T6 which used three different sources of N fertilizers if compared with T1 that used only one N fertilizer (urea). Thus, using local sources of organic fertilizer has the potential to increase rice production and give smallholders opportunities to enhance conventional fertilizer efficiency with improved plant performance and soil management [44].

### 4.2. The Effect of Combined Manure and Urea Treatments on Nutrient Uptake

Nutrient uptake is an important parameter in determining the effects of applied nutrients on crops. Therefore, knowledge of this index is fundamental to improving nutrient management strategies. Research related to nutrient uptake is useful to develop best management practices and to produce high yields while minimizing nutrient losses and costs associated with nutrient fertilization [45]. Inefficient nutrient use is one of the most limiting factors, considering the need to increase rice productivity.

Nitrogen uptake ratio was significantly different among the treatments considering that T6, T5, and T4 had a greater effect on N uptake than did T1, T2 and T3. These findings were consistent with previous studies where combined fertilizers increased N uptake and N use efficiency [31]. Combined fertilizers, where organic manure is present, seem to be able to improve soil physico-chemical properties, promoting more favourable growth of rice plant roots.

Higher N uptake in roots, straw and grain in the presence of combined fertilizers in contrast to that in single fertilizers (manure or urea) is corroborated by Ming-gang et al. [46]. Sahu et al. [45] emphasized that higher N uptake in association with manure is likely to due to solubilization of native nutrients, chelating complex forms of intermediate organic molecules produced during decomposition of the added manures, and the resulting mobilization and accumulation of different nutrients in plant parts. Higher N uptake in roots, straw and grain in the presence of combined fertilizers in contrast to that in single fertilizers (manure or urea) is corroborated by Ming-gang et al. [46]. Sahu et al. [45] emphasised

that higher N uptake in association with manure is likely to due to solubilisation of native nutrients, chelating complex forms of intermediate organic molecules produced during decomposition of the added manures, and the resulting mobilisation and accumulation of different nutrients in plant parts.

The combination of manure and urea fertilizer has a significant effect on P uptake with the highest registered in T5. According to Ming-gang et al. [46], P uptake is highly dependent on the type of fertilizer application, rice varieties and nutrient management. There was a great increase in total P uptake in the rice crop in the treatments with integrated application of manure and urea fertilizers compared to the use of a single fertilizer or the control. Similar results were observed in a study performed by Mitran et al. [47] who observed the highest P uptake in treatments amended with combined fertilizers. According to Kumar et al. [48], the increased P uptake might be due to organic materials forming chelates with $Al^{3+}$(aluminium) and $Fe^{3+}$ leading to more P available for plant uptake, particularly in soils with low P-fixing capacity.

Application of urea along with cattle and poultry manure (T4, T5 and T6) led to the best results for total K uptake in straw, grain and roots. This may be due to a higher availability of K in fertilizers and manure materials or a good spread of roots, resulting in greater K absorption [47]. Thus, the combined manure and urea fertilizer might be the best solution for availability of N, P and K.

Most smallholder farmers face significant cash constraints, that hamper making the decision of taking risks in adopting innovative technology related to crop production. In this context, economic assessment is important to determine the feasibility of proposed solutions. We evaluated the cost of fertilizers, the cost per ton of rice produced, the monetary income and finally, the profit. The economic returns were improved when manure and urea were combined (T4, T5 and T6). The economic returns for T2 and T3 appear not to be sustainable.

Considering that there is significant livestock production in this region, the use of only manure (T2 and T3) or a 50–50 mixture of cattle and poultry manure (not tested in this study) could be more profitable for smallholders. We reached the conclusion that N was more readily available from poultry manure than from cow manure, with slower N release in the latter. This can explain the better effect of the combination of these two manures. The use of manure is economically viable and therefore can be extended to farmers for adoption. The use of poultry manure, or even cattle manure, in irrigated systems also has the potential to increase returns on investments by smallholder farmers, e.g., in the purchase of chemical fertilizer.

## 5. Conclusions

The results observed in this study suggest that integrated use of manure and synthetic urea N sources is more efficient than isolated application of urea or animal manure for rice growth and yields as well as soil quality. The premise in this study was that solutions proposed here should not lead to any increase in production costs and should contribute to a significant increase in rice yields even for farmers with low or no capacity to invest in chemical fertilizers.

The combination of urea, beef cattle manure and poultry litter at a rate of (40%:30%:30%) produced good results and is thus recommended to smallholder farmers for better optimum crop production on the Chókwè Irrigation Scheme. Consequently, the smallholders should focus on manure combined with synthetic urea, which shows better results in response and economic evaluation. This study recommends more training be provided to smallholders to optimize alternatives and use of available resources such as animal manure fertilizer. This will help overcome difficulties in fertilization and contribute to enhance sustainable production and economic income in the region, also following a circular economy approach. Monitoring, however, is also advised to avoid any long-term environmental or health impacts that may occur.

**Author Contributions:** Conceptualization, F.I. and D.F.; methodology, F.I., A.N., and D.F.; formal analysis, F.I. and D.F.; investigation, F.I.; writing—original draft preparation. F.I.; writing—review and editing. F.I., A.N., and D.F.; supervision. D.F. and A.N. All authors have read and agreed to the published version of the manuscript.

**Funding:** This research was funded by the Fundação para a Ciência e Tecnologia (FCT) of the Portuguese Government through the grants to F.I. (SFRH/BD/128098/2016) and through the Linking Landscape, Environment, Agriculture and Food (LEAF) Research Unit (UID/AGR/04129/2020).

**Institutional Review Board Statement:** Not applicable.

**Informed Consent Statement:** Not applicable.

**Data Availability Statement:** Data sharing not applicable.

**Acknowledgments:** The authors are grateful to the International Rice Research Institute for providing the field area to conduct this experimental research. The authors also thank all smallholders from the Chókwè Irrigation Scheme in Gaza Province, Mozambique, for their great support. The authors thank Paula Alves for manuscript checking and English grammar correction, Alexandre Quetane for practical and helpful contribution during the fieldwork and finally the academic editor Arno Rosemarin for the English revision of the manuscript.

**Conflicts of Interest:** The authors declare no conflict of interest.

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
