# Peer review of "New Fertilizer Strategies Combining Manure and Urea for Improved Rice Growth in Mozambique"

_agronomy, doi:10.3390/agronomy11040783_

Round 1

Reviewer 1 Report

The topic of this work is interesting to researchers in relevant fields, and especially benefits to the studying region’s sustainable development.  The experiments were soundly fitting the goals. I have a few concerns on terminology and editorial consistency. Citation of a recent book and/or relevant chapters (Animal Manure: Production, Characteristics, Environmental Concerns and Management. ASA Special Publication 67. ASA and SSSA, Madison, WI) may enhance your introduction and discussion, and so that greater impacts of your work.

L22 and throughout the manuscript. Can you be more specific of the type of cattle manure as beef manure or dairy manure (Animal manure chapter 4)?

Also make sure if you really used “poultry manure”. In most “real world” poultry farming, the waste is “poultry litter” which is different from “poultry manure” (refer to their definitions in Animal manure chapter 1).

L32. “soil management and best practices” is confusing. Does it  mean “best soil management practices”?

L47 and L52, and other places. Be consistent to present the  values with one format of units (ton/ha vs. ton ha-1). By the way, ton may be changed to Mg.

L171. In table 2. Soil column. Total N is much less than NH4-N or NO3-N? Please also clarify if all these data are dry matter or fresh mass-based.

L212 Table 3. Better to convent data from “g.m-2” to “kg ha-1).

L228. Change “.” To “,”.

L303. Figure 1. X-axis dates are marked not as MM/DD/YYYY.

L307. No figure 2 can be seen excerpt to the title.

L371. “per hill-1” seems not right.

L415. Table 4. It seems not correct to show no effect by a downward arrow, which is more appropriate for decreasing trend. As a matter of fact, the table makes not much sense. Why not present quantitative data (% change)? Also give the p levels )0.05, 0.01 or 0.001) if you say significant.  

L709 and in previous tables. Please use terms accurately. Such  “cattle and poultry” should be “cattle manure and poultry manure”.

Author Response

Reviewer # 1-Remarks

  1. L22 and throughout the manuscript. Can you be more specific of the type of cattle manure as beef manure or dairy manure (Animal manure chapter 4)? Also make sure if you really used “poultry manure”. In most “real world” poultry farming, the waste is “poultry litter” which is different from “poultry manure” (refer to their definitions in Animal manure chapter 1).

Thanks, it has been revised. Based in housing systems and purpose of livestock in this specific region of Mozambique, the material used in the experiments is beef manure and poultry litter (Lines 22-25).   

  1. “soil management and best practices” is confusing. Does it mean “best soil management practices”?

Thanks for this important remark, it was already revised. The real objective of this work is focused on best practices for the smallholders, so we can address best soil management practices. Based in the statement that soil management practices are fundamental to achieve best crop performance (Line 32).

  1. L47 and L52, and other places. Be consistent to present the values with one format of units (ton/ha vs. ton ha-1). By the way, ton may be changed to Mg.

All the document has been revised regarding this issue. It has been changed to Mg ha-1 (Lines 47-53).

  1. In table 2. Soil column. Total N is much less than NH4-N or NO3-N? Please also clarify if all these data are dry matter or fresh mass-based.

It has been revised. The data presented in table 2 are on a dry matter basis. There was an mistake related with the conversion, the total N is not less than NH4-N even NO3-N, the real value has been corrected (Lines 174-181).

  1. L212 Table 3. Better to convent data from “g.m-2” to “kg ha-1).

It has been revised (Line 226).

  1. Change “.” To “,”.

It has been corrected (Line 222).

  1. Figure 1. X-axis dates are marked not as MM/DD/YYYY.

It has been revised (Line 296).

  1. No figure 2 can be seen excerpt to the title.

It was a mistake. The figure 2 has been included in the manuscript (Line 302).

  1. “per hill-1” seems not right.

The correct term is per panicle and not per hill. It was a mistake. It has been revised (Line 376).

  1. Table 4. It seems not correct to show no effect by a downward arrow, which is more appropriate for decreasing trend. As a matter of fact, the table makes not much sense. Why not present quantitative data (% change)? Also give the p levels (0.05, 0.01 or 0.001) if you say significant.  

Thanks for this important remark, it has been revised. We recognize that this was an important remark, and we would like to clarify that the idea was to assess the increase in the parameters, so we added now some new statements with the % change according to your remarks (Lines 417-426).

  1. L709 and in previous tables. Please use terms accurately. Such “cattle and poultry” should be “cattle manure and poultry manure”.

It has been corrected in all documents (Lines 775).

Reviewer 2 Report

The manuscript reports the results of an experiment of rice production with different fertilisation strategies. Although the experiment itself is not new, the location and the peculiarity of the area make the results interesting and the related consideration of practical use.  

The manuscript is well written and the methodology is clearly explained. The results are well presented and the discussion well conducted. I have just some minor comments to the manuscript.

Line 71-72 – For sure the fertilisation is a limiting factor but also water availability. The latter might be better addressed.

Line 122-123 and 127 – There are some contradiction between annual precipitation (500-700 mm) and total rainfall in the experimental period (3371 mm).

Line 140 – Table 1 – The data should be restricted to the  experimental period

Line 172 – Table 2 – As they values are experessed on dry matter basis, it would be useful to add the dry matter content. The poultry manure has a relative low share of mineral nitrogen. Is it a litter based manure?

Line 260 – I missed the harvesting date

Line 267-269 – The vegetative growth and maximum height are not always a good indicator of the yields.

Line 272 – I would consider equal the treatments with no significant differences, also if the means are in that order.

Line 307 – Figure 2 is missed in the pdf

Line 319-321 – If you are not discussing the values here, also this sentence can be moved to discussion

Line 379-380 – From the statistical analysis it results that T1, T4, T5 and T6 are not different.

Line 415 - Table 4 – For a better readability you can make the treatment more explicit adding the explanation in the caption or directly in the table.

Table 5,  6 and 7 – These data are interesting but consider to move some of them as supplementary material

Line 594 – T1, T4, T5 and T6 have the same yield.

Line 550 ff – The economic evaluation is a key factor in the fertilisation strategies. I think the price of rice should be considered for a more complete evaluation and to make a cost/benefit assessment.

Line 652 ff – The discussion of the combination of different fertilisers is really interesting. In my opinion should include also economic aspects and some different scenario in order to be more effective.

A cost-benefit assessment should be discussed. For example, the yield increase in T4, T5 and T6 might not be sustainable compared with T2 and T3 from the economic point of view.

Furthermore a possible strategy with 50% cattle and 50% poultry, although not tested in the experiment, might be a possible solution when the available money to buy urea are not available and probably the investment will not be remunerative.

Thus at least two scenarios might be considered: one with urea (T6) and one with organic fertiliser only. The first one when urea is available (for different reasons) and the second when it is not.

Some discussion about the different forms of nitrogen in the organic fertilisers will also help. Poultry manure should have more ready available nitrogen and cow manure should have nitrogen with slower release. This can explain the better effect of the combination of the two.

Line 702 ff – Conclusions – I would include some of the previous considerations also in this section enforcing the need of a circular economy in this type of production, that should not be depending from external input of mineral fertilisers but should enhance the use of available nutrient sources.

Author Response

Reviewer # 2 -Remarks

  1. Line 71-72 – For sure the fertilisation is a limiting factor but also water availability. The latter might be better addressed.

Thanks, it has been revised accordingly (Lines 72-75). Furthermore, water is another vital issue that constrains rice production, namely water access and additionally weak water management capacity in the fields to guarantee adequate supply in the necessary time [10].

  1. Line 122-123 and 127 – There are some contradictions between annual precipitation (500-700 mm) and total rainfall in the experimental period (3371 mm).

The first annual precipitation mentioned is according to the literature reviews that we had consulted. The total rainfall (3371 mm) was found during the experimental period that we conducted in the experimental field. These data were obtained by the local meteorological station, which gently gave us the data.

  1. Line 140 – Table 1 – The data should be restricted to the experimental period.

It has been removed (Lines 143-145).

  1. Line 172 – Table 2 – As they values are expressed on dry matter basis, it would be useful to add the dry matter content. The poultry manure has a relative low share of mineral nitrogen. Is it a litter based manure?

The values of dry matter content were added. The material used in this experiment is poultry litter (Lines 174).

  1. Line 260 – I missed the harvesting date.

The harvesting time was on 17th May 2019 (Line 223-224).

  1. Line 267-269 – The vegetative growth and maximum height are not always a good indicator of the yields.

 Thank you for the contribution. We would like to highlight that the idea was to evaluate the performance of rice plant under the treatments studied in this experiment. In accordance with the previous studies carried out by Moe et al. 2019 where they evaluated the effect of organic and inorganic fertilizers on growth characteristics.

  1. Line 272 – I would consider equal the treatments with no significant differences, also if the means are in that order.

Thank you very much for your suggestion. The values are in the order mentioned on document with no significant differences (Line 267).

  1. Line 307 – Figure 2 is missed in the pdf

It was a mistake. The figure 2 has been included in the manuscript (Line 302).

  1. Line 319-321 – If you are not discussing the values here, also this sentence can be moved to discussion.

The sentence has been moved to the discussion (Line 624-626).  

  1. Line 379-380 – From the statistical analysis it results that T1, T4, T5 and T6 are not different.

Thanks. It has been revised accordingly (Line 384- 385).

  1. Line 415 - Table 4 – For a better readability you can make the treatment more explicit adding the explanation in the caption or directly in the table

Thanks for this important remark, it has been revised (Line 417- 426).

  1. Table 5, 6 and 7 – These data are interesting but consider moving some of them as supplementary material.

The data presented in these tables are relevant for the discussion and attending that the total number of tables is not high, we prefer to keep it in the main manuscript.

  1. Line 594 – T1, T4, T5 and T6 have the same yield.

It has been revised (Line 633-634).

  1. Line 550 ff – The economic evaluation is a key factor in the fertilisation strategies. I think the price of rice should be considered for a more complete evaluation and to make a cost/benefit assessment.

Thanks for this important remark, it has been revised. To calculate the monetary yield of each treatment, the values of the weight of the rice in tonnes per hectare were used. Then, these values were multiplied by the sale price of rice to the smallholders (0,20€ / kg). The difference between monetary income and the cost of production was calculated as /resulted as a profit (Line 578-582).

  1. Line 652 ff – The discussion of the combination of different fertilisers is really interesting. In my opinion should include also economic aspects and some different scenario in order to be more effective. A cost-benefit assessment should be discussed. For example, the yield increase in T4, T5 and T6 might not be sustainable compared with T2 and T3 from the economic point of view. Furthermore, a possible strategy with 50% cattle and 50% poultry, although not tested in the experiment, might be a possible solution when the available money to buy urea are not available and probably the investment will not be remunerative. Thus at least two scenarios might be considered: one with urea (T6) and one with organic fertiliser only. The first one when urea is available (for different reasons) and the second when it is not. Some discussion about the different forms of nitrogen in the organic fertilisers will also help. Poultry manure should have more readily available nitrogen and cow manure should have nitrogen with slower release. This can explain the better effect of the combination of the two.

It has been revised accordingly (Lines 739-759). We emphasized more the economical assessment caried out in the study.

  1. Line 702 ff – Conclusions – I would include some of the previous considerations also in this section enforcing the need of a circular economy in this type of production, that should not be depending on external input of mineral fertilisers but should enhance the use of available nutrient sources.

We added the idea to emphasize the importance of circular economy. This study recommends more training and training should be provided to the smallholders allowing them the chance of optimizing other alternatives and available resources, such as organic manure to overcome the difficulties related to fertilization and contribute to enhance sus-tainability production and economic income in the region, following the circular economy basis (Lines 780-786).